# Comment on Laming et al. The Curse of Conservation: Empirical Evidence Demonstrating That Changes in Land-Use Legislation Drove Catastrophic Bushfires in Southeast Australia. *Fire* 2022, *5*, 175

**Ian Penna**

Independent Researcher, P.O. Box 89, Meredith, VIC 3333, Australia; ipenna@bigpond.com

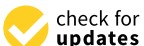



## 1. Introduction

In 1970, the Victorian state government in Australia established the Land Conservation Council (LCC) to study the state's publicly-owned land and make recommendations for its use [1]. Amongst its recommendations, the Council proposed new parks and reserves and confirmed which land could be used for various commercial activities, such as logging [2]. The possible consequences of its recommendations for relevant state government departments and public land management became politically contentious issues [3].

The impact of the Council's recommendations on fire management in eastern Victoria is the focus of the research effort described in 'The Curse of Conservation' [4]. The authors (Laming et al.) state that their basic objective was "to investigate how changing management approaches since the early 1900s in response to conservation/wilderness-inspired legislation influenced fire regimes in part of southeast Australia prone to catastrophic bushfires" [4].

To achieve their objective, Laming et al. attempted to relate the results of an analysis of one sediment core taken from a small lagoon adjacent to the Snowy River in eastern Victoria to their interpretation of the LCC's activities and its governing legislation. In particular, the authors argued that their analysis provided empirical evidence that the implementation of Victoria's Land Conservation Act: (i) caused an immediate increase in serious bushfires in that area; and, (ii) is implicated as the 'root cause' of the 2019–20 bushfires in southeast Australia [4].

This argument is flawed: it is based on factual mistakes and a poor research method.

## 2. Laming et al.'s Argument

The foundation of the authors' argument includes a variety of incorrect or unjustified statements about the Land Conservation Act and the Land Conservation Council that mislead the reader about the operation of the Act and the Council, as well as their potential and actual influence on fire management and impacts at the research site and surrounding region. In particular:

- The authors say that the Land Conservation Act of 1970 "prohibited burning by settler land holders in an effort to protect natural landscapes" [4]. This is wrong. The Act contains no such wording; it simply established the Land Conservation Council (LCC) [1].
- The authors blamed the Land Conservation Act/Council for the increase in fire activity at the research site from 1970 by incorrectly claiming that "settler mimicry burning" was banned in this area at this time [4]. However, the Act did not do this [1], and the LCC did not make any recommendations for public land in any study area in 1970. Its final recommendations for public land containing the research site were published in 1983 [5], and so were implemented by the government after this time.

- The authors state in their Abstract that "Our data demonstrate that catastrophic bushfires first impacted the local area immediately following the prohibition of settler burning in 1970, … … ." [4], and make other similar statements in the main text. However, information from the Victorian government's environment department shows no catastrophic bushfires impacted the local area immediately after 1970. Also, there were bushfires in the vicinity of the research site and Buchan prior to 1970. See Figure 1: Balley Hooley Campground fire history 1960 to 1990 [6]; the research site is adjacent to the Balley Hooley Campground on the Snowy River.

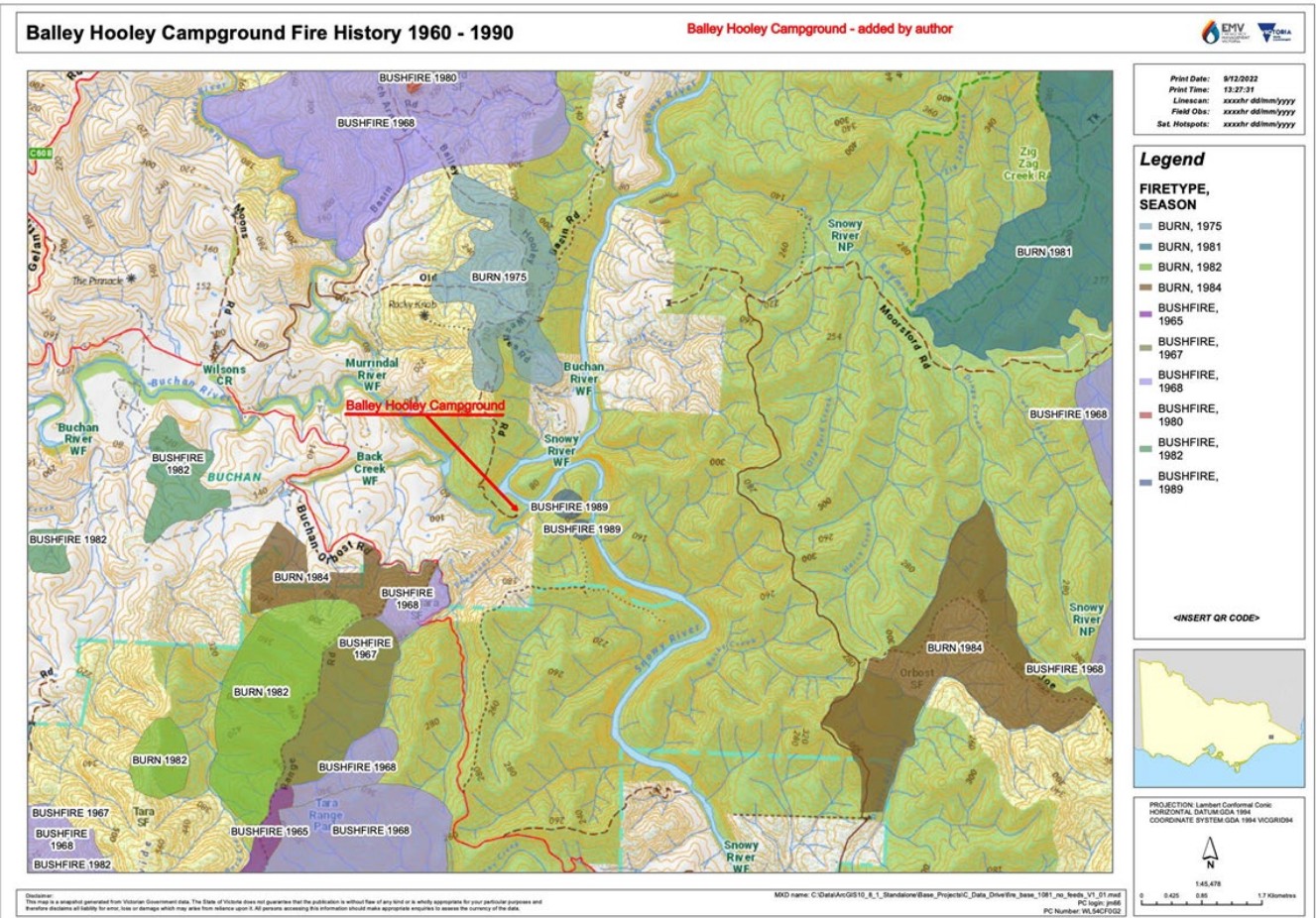

**Figure 1.** Balley Hooley Campground fire history 1960 to 1990 (DELWP 2022).

Thus, the foundation of Laming et al.'s primary argument is unsound.

## 3. Unjustified Generalisation of Research Findings

Laming et al. based their argument on the results of an analysis of one core sample from one research site in the west of East Gippsland; they then extrapolated the results across East Gippsland. This is an illegitimate behaviour, as an anonymous reviewer of the original manuscript of that study explained: "The limitations of sampling only one core sample, in one locality, should also be explicitly addressed. The results for one local area cannot be extrapolated to an entire state area" [7].

Further complicating this mistake, Laming et al. chose as their only research site an environment unique in Australia. For the very widespread whole of landscape effects as claimed by Laming et al., the authors need to show that the changes occurred across a variety of sites throughout East Gippsland. Laming et al. only show that something occurred at one site in a unique environment that cannot represent the variety of conditions across southeast Australia.

### 4. An Alternative Explanation

The research site is in the Snowy River environment, the only riverine system in Australia that, prior to the late 1960s, had large annual flows and was dominated by peak annual flows from the Alpine spring snowmelt [8]. After 1967 when the Jindabyne dam in New South Wales was finished [9], the river's annual and spring snow melt flows were dramatically reduced by the Snowy Mountains Hydro-power Scheme to the extent of allowing new vegetation to colonise areas next to the river and increasing sand storage at bars and benches [10]. This time—the late 1960s to the early 1970s—corresponds with the time that Laming et al. argue that vegetation changes began at the research site. However, their paper does not consider the impact on their research site and sediment cores of the centuries-long unrestricted river flows or the approximately 55 years of diminished river flows after the construction of dams for the Snowy Hydro Scheme.

### 5. Conclusions

Laming et al.'s argument does not stand up to reasonable scrutiny. Supposed facts about the Land Conservation Act/Council at the basis of their paper's argument are inexplicably wrong and/or misleading. Also, the authors did not attempt to replicate their analysis across East Gippsland. Their paper does not provide empirical evidence that demonstrates changes in land-use legislation (i.e., the Land Conservation Act) drove catastrophic or any other bushfires in southeast Australia at any time.

There is an alternative and more appropriate explanation for the cause of the changes in Laming et al.'s sediment core that the authors do not address: the 'deep colonisation' of the Snowy River Catchment in Australia's high country for the Snowy Mountains Hydro-power Scheme at the expense of Indigenous human and related ecological communities [11].

**Conflicts of Interest:** The author declares that in 1979, he was a member of the Land Conservation Council for about 6 months as a replacement for a nominee of the then Conservation Council of Victoria. In the early 1980s, he worked on a legal action taken by the Australian Conservation Foundation to stop the implementation of the LCC's Ovens Softwood Plantation Zone Special Investigation Final Recommendations, which if implemented would have resulted in the destruction of about 10,000 ha of native forests in north-east Victoria for the establishment of softwood plantations.

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
