# Peer review of "Comment on Laming et al. The Curse of Conservation: Empirical Evidence Demonstrating That Changes in Land-Use Legislation Drove Catastrophic Bushfires in Southeast Australia. Fire 2022, 5, 175"

_fire, doi:10.3390/fire6070253_

Round 1

Reviewer 1 Report

My own concern with this paper is the need to ensure that more context is given at the start so that readers will know more about where this critique has come from though - a few sentences about the original paper and its thesis (that changing management left a massive fire legacy) needs to be stated (this makes the critique a stand-alone document. 

Reviewer 2 Report

I believe this comment on the paper by Laming et al. is an important contribution to debates about sustainable fire management, management of land for nature conservation and cultural burning generally.  The author makes three points that are relevant to this the Laming article: (a) error of fact regarding management law and policy; (b) inability to generalise from a single site; and (c) providing a credible alternative explanation to the observed changes in the vegetation reconstruction (hydroelectric impoundment affecting river flows).  I found the tone of the article appropriate (noting some stylistic changes below).  

My only criticism is the layout of the paper, main headings (Introduction and Conclusion should be in bold and numbered, and subheading should be in italics and expressed more succinctly: The Curse of Conservation’s main argument, (better ‘Laming et al. argument’); A failure in the research method (better ‘Unjustified generalisation of research findings’); line 51 paragraph could have subheading ‘An alternative explanation’. In this section the author could propose that this hypothesis could eb tested on other rivers in the region that were not dammed such as the Mitchell River.

Reviewer 3 Report

This is an excellent and concise summary of one of the central issues with Laming et al, and I commend the author for their clarity. 

The argument in the paper is straightforward: Laming et al claimed that a piece of legislation had a given effect, when in reality the legislation was unrelated to the effect they describe. Further to this, the authors point out that Laming et al make an argument for a broad landscape based entirely on a single point, measured in a unique environment subject to a major change that occurred at the time in question. This argument is devastating to the Laming et al case.

It cannot be argued in defence of Laming et al that the acts achieved the cessation of burning through indirect means. Laming et al stated that the conservation acts "advocated for a complete removal of human agency" - a demonstrably false statement. Numerous reviews demonstrate that in reality, prescribed burning greatly increased in Victoria during this period - largely in lands established as part of the reserve system advocated by the act.

I have 2 small edits.
Lines 32-33: It would be good to reference this statement, even with a link to the spatial dataset.
Line 54: Again, a reference for this important date wud be good
Line 71: I suggest replacing "Australian High Country" with "Snowy River Catchment", as many readers may be unaware of the connection.

This is an important response and needs to be published with some urgency. 
